# Frozen Language Model Helps ECG Zero-Shot Learning

**Jun Li** [1,*]                                                     LIJUN2020@MAILS.JLU.EDU.CN
**Che Liu** [2,3,*]                                                  CHE.LIU21@IMPERIAL.AC.UK
**Sibo Cheng** [3]                                                   SIBO.CHENG@IMPERIAL.AC.UK
**Rossella Arcucci** [2,3]                                           R.ARCUCCI@IMPERIAL.AC.UK
**Shenda Hong** [4,5,†]                                              HONGSHENDA@PKU.EDU.CN

[1] *College of Electronic Science and Engineering, Jilin University, Changchun, China*

[2] *Department of Earth Science and Engineering, Imperial College London, SW7 2AZ, UK*

[3] *Data Science Institute, Department of computing, Imperial College London, SW7 2AZ, UK*

[4] *National Institute of Health Data Science, Peking University, Beijing, China*

[5] *Institute of Medical Technology, Health Science Center of Peking University, Beijing, China*

**Editors:** Accepted for publication at MIDL 2023

## Abstract

The electrocardiogram (ECG) is one of the most commonly used non-invasive, convenient medical monitoring tools that assist in the clinical diagnosis of heart diseases. Recently, deep learning (DL) techniques, particularly self-supervised learning (SSL), have demonstrated great potential in the classification of ECG. SSL pre-training has achieved competitive performance with only a small amount of annotated data after fine-tuning. However, current SSL methods rely on the availability of annotated data and are unable to predict labels not existing in fine-tuning datasets. To address this challenge, we propose **M**ultimodal **E**CG-**T**ext **S**elf-supervised pre-training (METS), **the first work** to utilize the auto-generated clinical reports to guide ECG SSL pre-training. We use a trainable ECG encoder and a frozen language model to embed paired ECG and automatically machine-generated clinical reports separately. The SSL aims to maximize the similarity between paired ECG and auto-generated report while minimize the similarity between ECG and other reports. In downstream classification tasks, METS achieves around 10% improvement in performance without using any annotated data via zero-shot classification, compared to other supervised and SSL baselines that rely on annotated data. Furthermore, METS achieves the highest recall and F1 scores on the MIT-BIH dataset, despite MIT-BIH containing different classes of ECG compared to the pre-trained dataset. The extensive experiments have demonstrated the advantages of using ECG-Text multimodal self-supervised learning in terms of generalizability, effectiveness, and efficiency.

**Keywords:** Multimodal self-supervised learning, Zero-shot learning, Language model, ECG, Signal processing

## 1. Introduction

The electrocardiogram (ECG) is a diagnostic tool that is widely used in clinical practice (Addison, 2005). In practice, the ECG is used to detect a wide range of cardiac conditions, including arrhythmias, heart attacks, and heart failure (Berkaya et al., 2018). Recently, deep

---

* Contributed equally

† Corresponding author

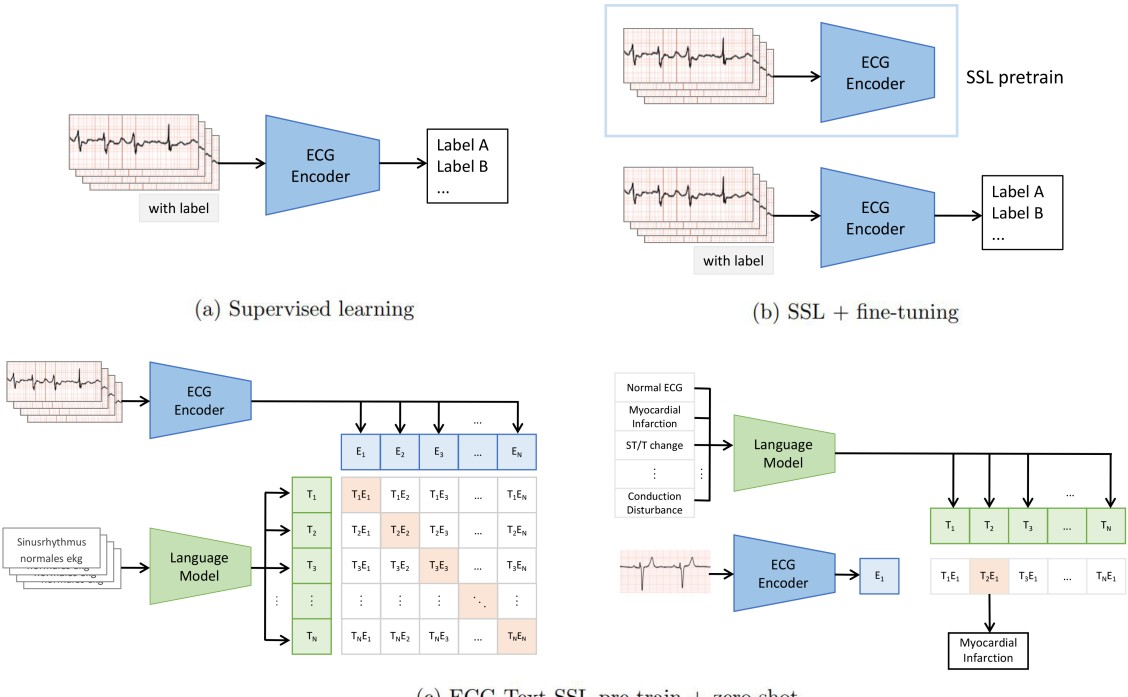

(a) Supervised learning

(b) SSL + fine-tuning

(c) ECG-Text SSL pre-train + zero-shot

Figure 1: (a) denotes a supervised learning method for ECG. (b) denotes a common self-supervised learning method for ECG, i.e. pre-training followed by fine-tuning. (c) denotes a self-supervised learning method for multimodal ECG-Text. Zero-shot classification is performed after pre-training is completed.

learning (DL) methods have shown promising results in classifying ECG data (Ebrahimi et al., 2020; Tripathy et al., 2019). DL models, such as convolutional neural networks (CNNs) and recurrent neural networks (RNNs), have been shown to be highly accurate in classifying ECG for a variety of cardiac conditions (Baloglu et al., 2019; Singh et al., 2018; Xu et al., 2020). However, training DL models in a supervised manner (see Figure 1 (a)) often requires a large number of high-quality labels to obtain strong generalization performance (Ebrahimi et al., 2020). In addition, some forms of ECG, such as ST-elevation myocardial infarction, are difficult to detect and often require manual interpretation of the ECG by trained cardiologists (Ayer and Terkelsen, 2014). This work requires a huge effort which is costly and laborious.

Currently, self-supervised learning (SSL) has achieved impressive performance on datasets containing a small number of annotations, which provides a promising solution for unanno-tated ECG data (Jaiswal et al., 2020; Chou et al., 2020). It allows models to mine useful representations of ECG and can be widely used for various downstream tasks such as ab-normality detection and arrhythmia classification (Lan et al., 2022; Mehari and Strodthoff, 2022). Nevertheless, existing ECG SSL methods still require a large amount of annotated data in order to fine-tune them for downstream tasks (see Figure 1 (b)). This require-

ment hinders the real-world application of ECG methods as some heart diseases are rare, which leads to problems with zero-shot learning. Zero-shot learning means that the model does not need any annotated samples for *unseen* categories (Socher et al., 2013). This is achieved by explicitly learning shared features from seen samples, and then generalizing them on unseen samples based on "descriptions" of the unseen categories' features (Xian et al., 2018; Pourpanah et al., 2020). Specifically, such "descriptions", are usually borrowed from external medical domain knowledge, textual ECG reports for example (see Figure 1 (c)). Zero-shot learning for ECG faces a number of challenges. The first challenge is the semantic gap, where ECG and text (automatically machine-generated ECG reports) are two heterogeneous modalities. ECG is long-term continuous numbers and text is relatively short-term discrete clinical terminologies (Krishnan and Sowmya Kamath, 2018). They are difficult to align and characterize each other (Liang et al., 2022). The second challenge is domain adaptation. Zero-shot learning model may be sensitive to unknown domains, making it difficult to adapt to new domains or unseen categories and not performing well for downstream tasks in zero-shot learning. The third challenge is scalability. Zero-shot learning models need to learn a large number of representations and apply them to downstream tasks, which leads to a large computational cost for the model (Wang et al., 2019). Recently, (Yamaç et al., 2022) and (Bhaskarpandit et al., 2022) have reached considerable results on ECG zero-shot classification tasks. However, they pre-trained the model with supervised learning, which indicates that their methods still require large-scale annotated ECG for the pre-training stage.

To fully utilize the unannotated data, CLIP (Radford et al., 2021) and ALIGN (Jia et al., 2021a) first implement multimodal SSL with two individual encoders and use zero-shot classification as the downstream task to evaluate SSL pre-trained model performance (Radford et al., 2021; Jia et al., 2021b). Florence (Yuan et al., 2021), LiT (Zhai et al., 2022), and ALBEF (Li et al., 2021) explore the potential of multimodal SSL on large-scale pre-training tasks (Yuan et al., 2021; Li et al., 2021; Zhai et al., 2022). Although recent works have achieved substantial progress on an image-text task, the medical signals-text, such as ECG, has not yet leveraged the benefits of the multimodal SSL.

To take advantage of multimodal SSL, we propose a novel method to do **M**ultimodal **E**CG-**T**EXT **S**SL pre-training (METS). The METS model takes the ECG and the corresponding reported text as input and feeds them into a multimodal comparative learning framework. The multimodal framework contains a language component and an ECG encoder to obtain embedding representations of the text and the ECG respectively. To make full use of the a priori clinical knowledge in the report text, the report text is fed into a large frozen language model with Resnet1d-18 as the backbone of the ECG encoder, both of which have a linear projection head to embed the text and ECG into the same dimension. Then, the similarity between ECG embedding and text embedding is computed to minimize the contrast learning loss and obtain a pre-trained model with rich medical knowledge. probabilities, which can be used to classify the various categories of ECG.

The main contributions of this paper are summarised as follows:

- Our proposed METS is **the first work** to apply a large language model for ECG SSL. The apriori clinical knowledge from the language model can be fully exploited to help generate ECG medical reports.

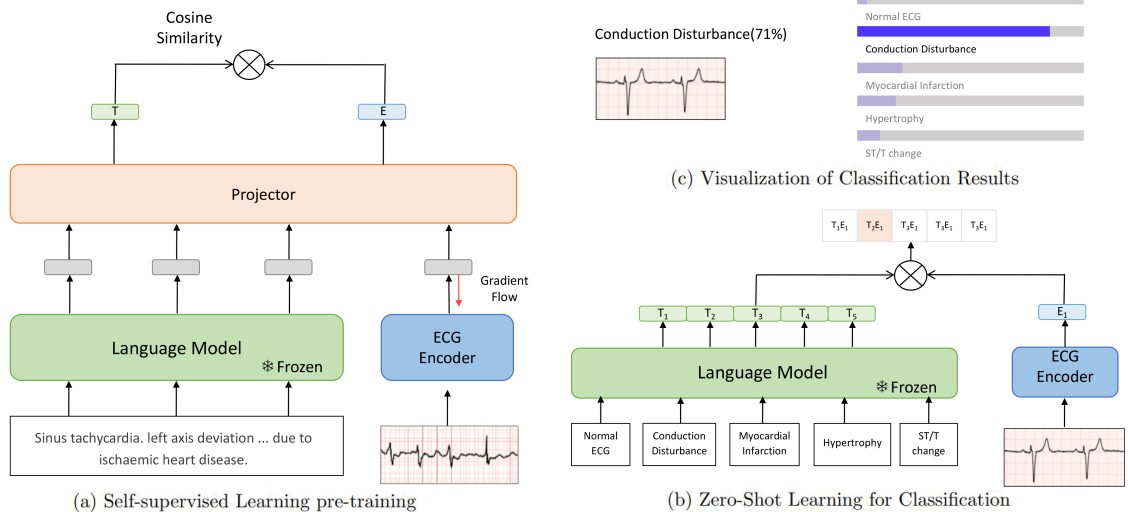

Figure 2: A framework for the METS approach. (a) shows a self-supervised pre-training approach. ECG-text pairs are fed into the model, and after comparative learning, the ECG encoder learns the parameters. (b) shows the zero-shot classification task. The corresponding labels are found by computing ECG and text similarity. (c) shows the visualization of the results of zero-shot classification.

- METS is independent of the annotated categories. Even if the external dataset categories are unseen, the classification can be done directly with zero-shot, unlike other SSL which require fine-tuning.

- Experiments demonstrate that METS can be adapted to any of the downstream tasks, e.g. form, rhythm, and superclass, without the need to fine-tune on different tasks. Besides, METS does not require any annotated data but can exceed the supervised and SSL methods of fine-tuning with small-scale annotated data.

## 2. Methods

In this section, we will demonstrate the details of METS. The framework is shown in Figure 2. METS consists of two components: Multimodal self-supervised pre-training (Section 2.1), and the zero-shot classification downstream task (Section 2.2).

### 2.1. Multimodal self-supervised pre-training

#### 2.1.1. Frozen Pre-trained Language Models

Our approach starts with a large pre-trained language model based on the transformer architecture. In order for the large language model to fully understand the report text, we extend the report text into the language model as a complete sentence input (Radford et al., 2021). Specifically, we construct a prompt template for the report **"The report of the ECG is**

**that {text}"**. We use a large clinical language model as the backbone of the text component. ClinicalBert has been pre-trained on all text from MIMIC III dataset (Alsentzer et al., 2019).

### 2.1.2. ECG ENCODER

Our ECG encoder $E_{ecg}$ is based on ResNet1d-18, which modifies the kernel of ResNet-18 from a 2D patch to a 1D stride, in order to obtain the deep ECG embeddings e (He et al., 2016; Hong et al., 2020). This process can be represented as follows: $\mathbf{e} = E_{ecg}(\mathbf{y})$, where $\mathbf{y}$ is the input of ECG. Then, a linear projection head $f_e$ maps raw embeddings to $\mathbf{e}_d \in \mathbb{R}^D$. The embedding dimension of the ECG encoder is set to be the same as the language model embedding dimension $d$ for comparison learning.

Inspired by (Tsimpoukelli et al., 2021), we freeze the parameters of the language model (LM) and use only paired ECG-text data from the PTB-XL dataset to update the parameters of the ECG encoder during SSL pre-training. This has the advantage of allowing the ECG encoder to learn rich prior clinical knowledge from the medical corpus, thus improving the generalization ability of the model. In addition, the parameters of the language model are frozen to reduce the significant computational cost of LM parameter updates.

### 2.1.3. MULTIMODAL CONTRASTIVE LEARNING

Following the multimodal contrast learning framework, we treat a pair of report text and ECG belonging to the same patient as a positive sample pair, while treating pairs of other patients' report text and that ECG as negative sample pairs. We maximize the contrast loss of different pairs $(\mathbf{t_i}, \mathbf{e_j})$ and minimize the contrast loss of the same pair $(\mathbf{t_i}, \mathbf{e_i})$ to improve the similarity of the same pair of samples. We first define the similarity between the representations $\mathbf{t}$ and $\mathbf{e}$ of two modalities in terms of cosine similarity, as shown in Equation 1.

$$\text{sim}(\mathbf{t}, \mathbf{e}) = \frac{t^\top \cdot e}{\|t\| \, \|e\|} \tag{1}$$

Then, we need to train two contrast loss functions. The first loss function is the ECG-to-text contrast loss for the $i^{th}$ pair, as shown in Equation 2.

$$\ell_{\mathbf{i}}^{(e \to t)} = -\log \frac{\exp\left(\text{sim}(\mathbf{t_i}, \mathbf{e_i})/\tau\right)}{\sum_{k=1}^{N} \exp\left(\text{sim}(\mathbf{t_i}, \mathbf{e_j})/\tau\right)} \tag{2}$$

The initialization of $\tau$ is set to 0.07.Similarly, the text-to-ECG contrast loss Equation 3 is represented as follows.

$$\ell_{\mathbf{i}}^{(t \to e)} = -\log \frac{\exp\left(\text{sim}(\mathbf{e_i}, \mathbf{t_i})/\tau\right)}{\sum_{k=1}^{N} \exp\left(\text{sim}(\mathbf{e_i}, \mathbf{t_j})/\tau\right)} \tag{3}$$

Finally, our training losses are calculated as the average combination of the two losses for all positive ECG-text pairs in each minibatch, as shown in Equation 4.

$$\mathcal{L} = \frac{1}{N} \sum_{i=1}^{N} \frac{\ell_{\mathbf{i}}^{(e \to t)} + \ell_{\mathbf{i}}^{(t \to e)}}{2} \tag{4}$$

## 2.2. Zero-Shot ECG Classification

In zero-shot classification, a segment of the ECG is used as input. To evaluate the zero-shot performance of the model on a multi-label classification task, we extend the discrete labels into full medical diagnostic statements and feed them into the language model to obtain embedding representations. Finally, the similarity between ECG embedding and text embedding is computed to obtain probabilities, which can be used to classify the various categories of ECG.

## 3. Experiments

### 3.1. Datasets

**PTB-XL**   We use the PTB-XL dataset to train the METS model (Wagner et al., 2020). The PTB-XL dataset contains 21,837 clinical 12-lead ECG of 10 seconds duration from 18,885 patients, where each ECG segment is paired with the corresponding ECG reports. The reports are generated by the machine and only describe the ECG without final diagnosis. The original ECG reports were written in 70.89% German, 27.9% English, and 1.21% Swedish, and were converted into structured SCP-ECG statements. The statement sets were assigned to three non-reciprocal categories: diagnosis, form, and rhythm. Specifically, the dataset consisted of 71 different statements, broken down into 44 diagnostic statements, 12 rhythmic statements, and 15 formal statements. For the diagnostic labels were divided into 5 superclasses and 24 subclasses. In the current experiments, we focused on investigating ECG-text pairs without using any other labels. Following the experimental setup in  (Huang et al., 2021; Wang et al., 2022), we extracted a multiclass classification dataset, PTB-XL test set, from the test set split.

**PTB-XL Test Set**   The original ECG in the PTB-XL dataset is multi-labeled with diagnostic, form, and rhythm. In zero-shot downstream task classification, we need to calculate the similarity of ECG and text to find the most similar target, and multiple labels for a target can confuse the categories. Therefore, we produce diagnostic superclass, form, and rhythm test sets to complete the corresponding zero-shot downstream tasks. There are 1,000 samples on each test set. Details of the specific split test set are shown in figure 4.

**MIT-BIH Test Set**   We use the MIT-BIH dataset for testing to evaluate the performance of our pre-trained representation framework for classification on external datasets (Moody and Mark, 2001). Please note that we do not pre-train on the MIT-BIH dataset. Similarly, we produced an MIT-BIH test set following the segmentation method above. Details of the specific split test set are also shown in figure 4.

### 3.2. Implementation Details

The models for the transformer were taken from the transformer library (Wolf et al., 2020). We took a linear projection head with an output dimension of 128 and a temperature $\tau$ initialized to 0.07. The ECG encoder is optimized using Adam optimizer with a learning rate of 1e-3 and weight decay of 1e-3. We use 50 epochs and a batch size of 32 for pre-training and downstream tasks. The experiments were conducted using PyTorch 1.7 on NVIDIA GeForce RTX-3090 GPU, which took about 8 hours.

Table 1: *PTB-XL* result on superclass. % refers to fractions of labels used in the training data.

| Methods | Accuracy | Precision | Recall | F1 |
|---|---|---|---|---|
| Self-supervised | | | | |
| random - 5% | 0.581 | 0.438 | 0.421 | 0.429 |
| SimCLR - 5% | 0.648 | 0.545 | 0.443 | 0.485 |
| **METS - 0%** | **0.842** | **0.694** | **0.626** | **0.657** |
| Supervised | | | | |
| Resnet18 - 100% | 0.894 | 0.811 | 0.745 | 0.776 |

Table 2: *PTB-XL* result on form. % refers to fractions of labels used in the training data.

| Methods | Accuracy | Precision | Recall | F1 |
|---|---|---|---|---|
| Self-supervised | | | | |
| random - 5% | 0.603 | 0.364 | 0.342 | 0.351 |
| SimCLR - 5% | 0.660 | 0.446 | 0.471 | 0.456 |
| **METS - 0%** | **0.734** | **0.537** | **0.503** | **0.518** |
| Supervised | | | | |
| Resnet18 - 100% | 0.724 | 0.520 | 0.508 | 0.509 |

### 3.3. Baselines

To demonstrate the performance of the METS method, our approach is compared with the following baselines. (1) **ResNet-18** (He et al., 2016): We choose ResNet-18 for showing the performance of small data fraction fine-tune. (2) **SimCLR** (Chen et al., 2020): A self-supervised contrast learning model that achieves good performance in SSL. We compare it with ECG SSL. The temperature parameter of SimCLR is set to 0.1. For all SSL methods above, we use 5% data for fine-tuning. (3) **Supervised** (He et al., 2016): We train ResNet1d-18 in a supervised manner, in order to compare the learning performance of our method with that of a fully supervised.

### 3.4. Results and Discussion

In this experiment, we assessed the ECG classification using the commonly used metrics: Accuracy, Precision, Recall, F1. We first performed a zero-shot classification of the PTB-XL Test set for the diagnostic superclass. We illustrate the classification results for the diagnostic superclasses in Table 1. It can be found that our method outperforms all other SSL methods with comparable performance to supervised training. For example, compared to SimCLR, accuracy and F1 are improved by 11% and 4%, respectively. In contrast, METS far outperforms other SSL methods in the form classification results presented in Table 2 and achieves better performance than supervised learning in accuracy, precision, and F1. As shown in Table 3, METS also achieves good performance for the rhythm classification

Table 3: *PTB-XL* result on rhythm. % refers to fractions of labels used in the training data.

| Methods | Accuracy | Precision | Recall | F1 |
|---|---|---|---|---|
| Self-supervised | | | | |
| random - 5% | 0.627 | 0.435 | 0.442 | 0.438 |
| SimCLR - 5% | 0.697 | 0.516 | 0.565 | 0.549 |
| **METS - 0%** | **0.746** | **0.576** | **0.612** | **0.593** |
| Supervised | | | | |
| Resnet18 - 100% | 0.790 | 0.664 | 0.607 | 0.633 |

Table 4: *MIT-BIH* result. % refers to fractions of labels used in the training data.

| Methods | Accuracy | Precision | Recall | F1 |
|---|---|---|---|---|
| Self-supervised | | | | |
| random - 5% | 0.565 | 0.468 | 0.499 | 0.483 |
| SimCLR - 5% | 0.749 | 0.642 | 0.610 | 0.624 |
| **METS - 0%** | **0.794** | **0.680** | **0.735** | **0.706** |
| Supervised | | | | |
| Resnet18 - 100% | 0.836 | 0.697 | 0.712 | 0.704 |

task. Overall, our results in PTB-XL show that the representations learned by METS are more informative than those of other state-of-the-art SSL methods. This also demonstrates that reports containing a priori knowledge can improve performance on the metrics.

We evaluated the performance of METS in migration learning. Table 4 shows the performance comparison under cross-dataset testing. In general, METS outperforms other state-of-the-art methods and even outperforms supervised learning. Compared to Table 1, there is a significant improvement in F1 for METS. This suggests that the features learned by METS are robust and have the potential to be generalized to other data sources.

## 4. Conclusion

In this paper, we present METS that uses automatically generated clinical reports to guide ECG pre-training. We pre-train the ECG encoder by applying the rich medical knowledge from the frozen large language model to the report text. As a result, our approach is independent of the class of annotated data and can be directly migrated to any unseen database. We can also do classification directly with zero-shot, unlike other SSL methods that require fine-tuning. Our experiments demonstrate that METS can be adapted to various downstream tasks, e.g. form, rhythm, disease, and abnormality classification. This means that the METS approach is more effective and efficient.

## Acknowledgement

This work was supported by the National Natural Science Foundation of China (No.62102008).

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

## Appendix A. Frozen Language Model Details

The ClinicalBert language model trained on all MIMIC notes and was initialized from BioBERT. The model was trained using code from Google's BERT repository on a GeForce GTX TITAN X 12 GB GPU. Model parameters were initialized with BioBERT. The model was trained using a batch size of 32, a maximum sequence length of 128, and a learning rate of 5e-5 for pre-training language models. The models trained on all MIMIC notes were trained for 150,000 steps. The dup factor for duplicating input data with different masks was set to 5. Specifically, masked language model probability = 0.15 and max predictions per sequence = 20 (Alsentzer et al., 2019).

## Appendix B. Example texts of inputted Frozen Language Model

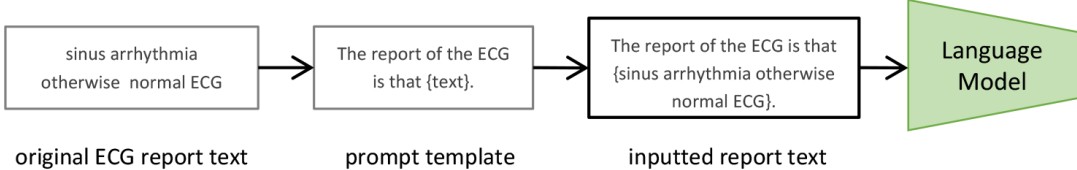

(a) A sample of inputted text report in pre-training SSL

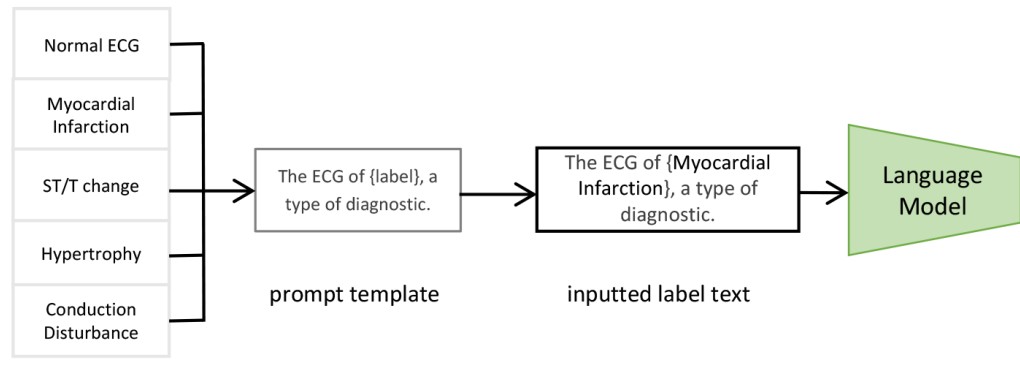

(b) Samples of inputted labels in zero-shot classification

Figure 3: Example texts of inputted Frozen Language Model.

Example texts of inputted Frozen Language Model are shown in figure 4.

First, during the pre-training process of self-supervised learning, in order for the large language model to understand the report text, we extend the report text into the language model as a complete sentence input. A sample inputted text report is shown below.

**"The report of the ECG is that {sinus arrhythmia otherwise normal ECG}"**.

Second, in zero-shot classification, we expand the labels into complete sentences for input into the frozen language model. This may be useful for specifying categories. Here are a few examples.

In the diagnostic classification task: **"The ECG of {X}, a type of diagnostic."** where X represents the different superclass diagnostic labels, e.g. Normal ECG, Conduction Disturbance, Myocardial Infarction, etc.

In the form classification task: **"The ECG of {Y}, a type of form."** where Y represents a different form label, such as Abnormal QRS, Non-diagnostic abnormalities, etc.

## Appendix C. Description of test sets.

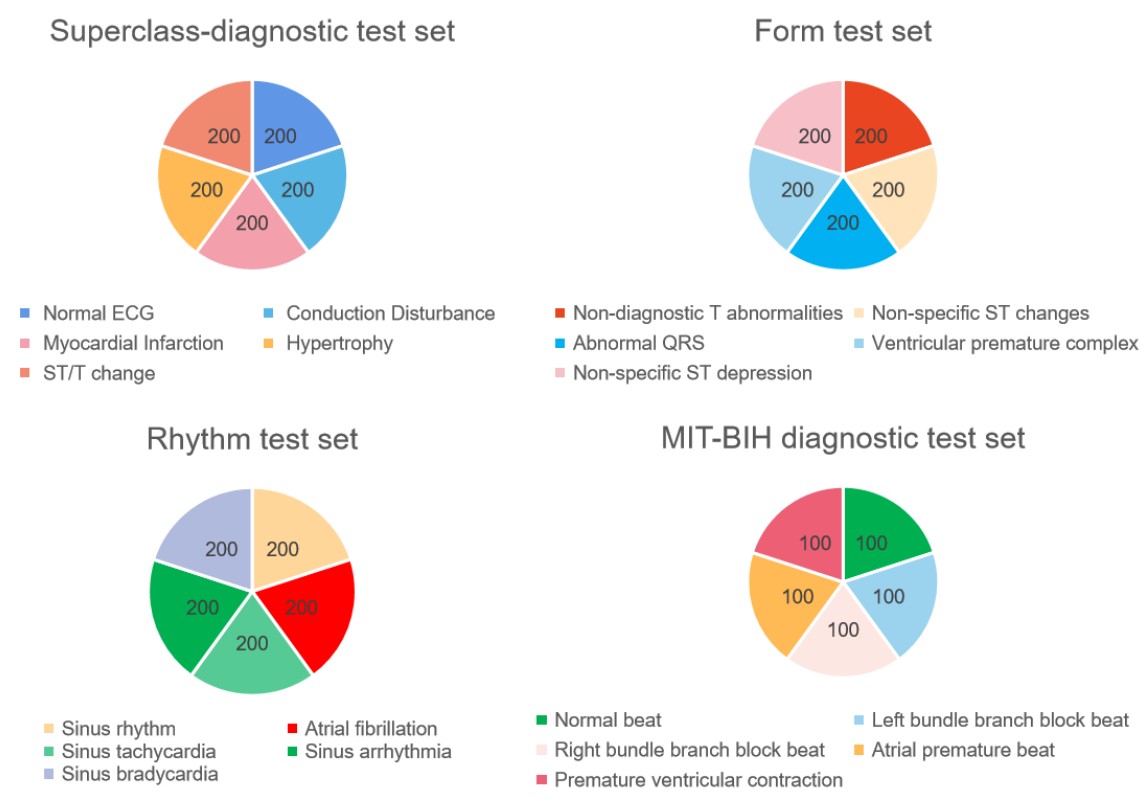

Figure 4: Description of test sets.

