# OpenReview forum: "Frozen Language Model Helps ECG Zero-Shot Learning"
_MIDL.io/2023/Conference — MIDL 2023 Oral_

### Official Review · Reviewer_b4Fg · 2023-02-03

**Confidence:** 5
**Preliminary Rating:** 4

**Summary:**

The authors used auto generated clinical reports and ECG signal data as multi-model pretraining in self-supervised learning.
The authors tested the results of hte self-supervised pretraining with zero shot learning in a different sets of class labels and demonstrated better results than supervised learning with fine tuning with liimted annotated data.

**Strengths:**

Self supervised learning is an important area in medical AI and it is interesting to see it's application in multi modal self supervised learning with text and images.

Used different languages of text interestingly.

**Weaknesses:**

It's unclear what is the actual output of the final model - diagnosis based on rhythms or disease classes?
It is unclear correlation between form, rhythm and diagnosis that could explain some of the results.
Unclear real world significance.

**Deanonymize Review:**

no

**Paper Type:**

validation/application paper

**Questions To Address In The Rebuttal:**

- should include examples of autogenerated clinical text - how are they generated and how good are they?
- should include examples of zero shot learning for classification
- should add in discussions how performances could be further improved

---

### Official Review · Reviewer_5YX7 · 2023-02-04

**Confidence:** 4
**Preliminary Rating:** 5
**Recommendation:** Oral

**Summary:**

In this work, the author present a novel approach to perform self-supervised learning in ECG analysis by incorporating textual health records into the pre-training pipeline. Thus, they propose to use a frozen large language model (LLM) and to use the similarity between the language description embedding and the ECG embedding as a self-supervised pretaining task.

**Strengths:**

The method seems to be novel and has huge potential!
The possibility to apply zero-shot learning for several different tasks by SSL pertaining with textual case reports is amazing, and the results are promising.
Congratulations to this nice work!

**Weaknesses:**

In general, the method and the experiments are sound and profound.
I am only missing some details on the exact architectures of the sub-networks. Maybe, a short paragraph in the appendix would be helpful.
Furthermore, Fig. 3 is way too small to grasp the details of the datasets. Please increase its size (maybe by shifting it to the appendix).


In general, I have concerns whether this work falls within the scope of MIDL since it has nothing to do with medical images. Thus, even though I strongly vote for an accept, I am unsure about the papers presentation as an oral/award given its topic. But this decision should be taken in the program board and my concerns are purely based on the scope and not on the manuscript.

**Deanonymize Review:**

no

**Detailed Comments:**

As mentioned above, I would mainly ask you to add some details about architectures to increase reproducibility. Furthermore, increasing image sizes or choosing different ways of visualizing the label distribution within the data sets might be helpful. Note that the appendix section is not limited in its size.

**Paper Type:**

methodological development

**Questions To Address In The Rebuttal:**

As mentioned above, I have very few concerns, and they are all of minor nature. I propose to rework a bit regarding the reproducibility.

Additionally, given the scope of MIDL conference, it would be nice to add some thoughts on the transferability to other domains, e.g. image analysis. Could the same method be adapted to work with images as well? Are there additional challenges?

---

### Official Review · Reviewer_KxjY · 2023-02-04

**Confidence:** 4
**Preliminary Rating:** 4
**Recommendation:** Oral

**Summary:**

This paper proposes a zero-shot classification model of ECG based on a self-supervised (SS) training procedure combining ECG and clinical report. The heterogeneous fusion of text and signal is performed by encoding separately the text and the ECG into distinct embeddings, then using the same linear projection to project them in the embedding space of same dimension and using a cosine similarity loss to train the model (with frozen language model) treating pairs of report text and ECG belonging to the same patient as a positive samples, while treating pairs of other patients’ report text and that ECG as negative samples. This architecture is shown to perform well in a zero-shot classification task, outperforming state-of-the art SS learning models, such as SimCLR, which require finetuning on a small dataset. The proposed model is also shown to generalize well to other pathologies that were not seen during the pretraining phase.

**Strengths:**

-The paper addresses two hot topics of the community concerning 1) the fusion of heterogeneous modalities (signal and text report) and 2) self-supervised learning.
-As far as I know (I am not expert in the domain of heterogeneous modality fusion), the paper presents some novel contributions in the field.
-The state-of-the art section is well documented with recent references
-The experimental study is well-conducted on two large public datasets.
-The proposed architecture is shown to perform well (including with unseen pathologies at inference) in a zero-shot classification task as well as outperform  state-of-the art SS learning models combined to finetuning.


**Weaknesses:**

-Description of the language model should be more detailed for non-expert MIDL attendees. It would be nice to add some example text reports inputted to the frozen language model, both during training and inference (e.g. In Appendix)

**Deanonymize Review:**

no

**Detailed Comments:**

-The authors should provide some discussion regarding the choice to freeze the language model parameters. Did they compare with the model trained end-to-end (including the language model)?

-In Section 2.2, please clarify “we extend the discrete labels into full medical diagnostic statements and feed them into the language model to obtain embedding representations.”

-In Section 3.1, please clarify the terms 'SCP-ECG statements' and 'utterance'.

**Paper Type:**

both

**Questions To Address In The Rebuttal:**

Please discuss the points i listed as 'weak' , regarding description of the language model. Also please address the questions of the 'comment' section, especially regarding the choice to freeze the language model.

----------------------------------------------------------------------------------------------
Dear Authors, Thank you for your reply to my comments and the revised version of the manuscript. I am pleased with your answers and the revised version of the manuscript.

---

### Meta-Review · Area_Chair_1SR2 · 2023-02-24

**Recommendation:** Accept (Oral)
**Confidence:** 4

**Metareview:**

The authors perform multi model fusion of ECG data and text data (corresponding generated reports) via contrastive learning. For zero-shot classification, first unseen labels are separately fed as text, then embeddings of each label are correlated with the embeddings of the corresponding ECG, and finally the classification is made as the label with maximum correlation with the ECG signal.
Overall, all reviewers are satisfied with the study as it is novel with very promising results. There were minor comments, which were addressed by the authors.